# Effects of *Litsea cubeba* Essential Oil Incorporated into Denture Soft Lining Materials

**DOI:** 10.3390/polym14163261

**Published:** 2022-08-10

**Authors:** Nichakorn Songsang, Chuchai Anunmana, Matsayapan Pudla, Trinuch Eiampongpaiboon

**Affiliations:** 1Residency Training Program, Department of Prosthodontics, Faculty of Dentistry, Mahidol University, Bangkok 10400, Thailand; 2Department of Prosthodontics, Faculty of Dentistry, Mahidol University, Bangkok 10400, Thailand; 3Department of Oral Microbiology, Faculty of Dentistry, Mahidol University, Bangkok 10400, Thailand

**Keywords:** denture liners, *Candida albicans*, *Streptococcus mutans*, hardness, cell survival

## Abstract

The antimicrobial properties, cell cytotoxicity and surface hardness of soft lining materials (GC soft liner, Viscogel and Coe comfort) incorporated with various concentrations of *Litsea cubeba* essential oil (*LC*EO) were evaluated. The minimum inhibitory concentrations of *LC*EO against *Candida albicans* and *Streptococcus mutans* were 1.25% *v*/*v* and 10% *v*/*v*, respectively. However, when *LC*EO was incorporated into the three soft lining materials (GC soft liner, Viscogel and Coe comfort), 10% *v*/*v* and 30% *v*/*v* of *LC*EO could inhibit the growth of *C. albicans* and *S. mutans*, respectively. The extracts of soft lining materials with 10% and 30% *v*/*v* *LC*EO, 2% chlorhexidine, 30% *v*/*v* nystatin and no additive were used for cytotoxicity tests on a human gingival fibroblast cell line. There was no significant difference in cell viability in all groups with additives compared to the no additive group (*p* > 0.05). Surface hardness increased significantly between 2 h and 7 day incubation times in all groups, including the controls (*p* < 0.05). A higher *LC*EO concentration had a dose-dependent effect on the surface hardness of all soft lining materials (*p* < 0.05). However, the surface hardness of materials with additive remained in accordance with ISO 10139-1. *LC*EO could be used as a natural product against oral pathogens, without having a negative impact on soft lining materials.

## 1. Introduction

Soft lining materials are used in various clinical treatments, such as relieving inflamed tissue, covering sharp atrophic ridges and improving denture fitting [1]. Soft lining materials are commonly used to relieve abused tissue and improve denture adaptation with antibiotic mouth rinse prescription in the case of denture stomatitis, a denture-induced oral infection among denture wearers [2]. Denture liners, on the other hand, are still prone to microbial accumulation, which typically includes *Candida albicans*, *Staphyloccocus aureus* and *Streptococcus mutans* [3]. Antimicrobial additives have been incorporated into soft lining materials as a drug delivery vehicle to improve the material properties and treatment effectiveness. Various substances, including nystatin, azole drugs, metallic oxide particles and natural products, have been chosen to combine into soft lining materials [4]. Due to the increase in antibiotic resistance and toxicity, alternative, naturally derived additives have become incredibly common.

*Litsea cubeba*, an evergreen tree from the *Lauraceae* family, is widely found in China, Japan and Southeast Asia. *Litsea cubeba* essential oil (*LC*EO) is extracted from various parts of the tree. It is reported that *LC*EO has the following pharmaceutical advantages: anti-inflammatory, anti-cancer and antimicrobial properties [5,6,7,8]. Chemical compounds vary depending on the part of tree; for example, citral, citronellal and limonene are the most common compounds in the roots and fruits, whilst β-phellandrene and β-terpinene are major compounds found in the stems and flowers [7,8]. In a previous study, a potent growth-inhibitory effect of *L. cubeba* on yeast and fungi (*Saccharomyces cerevisiae*, *C. albicans* and *Aspergillus nigus*) was revealed. In an agar disc diffusion assay, *L. cubeba* oil with citronellal as the main component completely inhibited this growth. The oil also affected the growth of Gram-positive bacteria (*Bacillus subtilis* and *S. aureus*), but it showed no effect on Gram-negative bacteria (*Pseudomonas aeruginosa* and *Escherichia coli*) [8]. Another study showed that citral had broad antimicrobial effects against Gram-negative and Gram-positive bacteria, fungi and protozoa. However, Gram-negative bacteria (*P. aeruginosa* and *Klebsiella pneumoniae*) are more resistant to citral than others [9].

Additionally, depending on the extraction procedure, traditional plants may have a variety of active chemical capabilities. In a previous work, the antioxidant capacity of *Ephedra foeminea* was changed by the use of various solvent extracts [10]. Another issue would be the drug form of an antibacterial agent. According to Khan et al., Cefuroxime axetil (CA), a semi-synthetic cephalosporin family of antibacterial antibiotics, has low oral bioavailability. To boost CA’s solubility and cellular absorption capacity, a lipid-based self-nanoemulsifying drug delivery system was developed [11].

Nevertheless, adding substances into soft lining materials may adversely affect the mechanical properties of the materials. Therefore, it is crucial to maintain the most favorable condition in which antimicrobial activity is improved without compromising the other properties. Surface hardness is one of the required properties of soft lining materials for clinical use. Moreover, the cytotoxic effect on human gingival fibroblast cells is of concern, and the cytotoxicity of soft lining materials is attributed to leachable products [12].

Therefore, the aim of this study was to determine the optimum concentration of *LC*EO that could be used as an antimicrobial additive incorporated into soft lining materials while not significantly altering the surface hardness of the materials.

## 2. Materials and Methods

### 2.1. Antimicrobial Additives

*LC*EO (Thai-China Flavours and Fragrances Industry Co., Ltd., Nonthaburi, Thailand) containing citral as an active component was obtained by the steam distillation extraction technique from fruits. Nystatin oral suspension 100,000 IU/mL or 33.3 mg/mL (Continental Pharm Co., Ltd., Bangkok, Thailand) was selected to represent a conventional antifungal drug against *C. albicans*. Moreover, 2% chlorhexidine gluconate mouthwash (Faculty of Dentistry, Mahidol university, Bangkok, Thailand) was used as a positive control against *S. mutans*.

### 2.2. Soft Lining Material

GC soft liner (GC Dental Products Corp., Tokyo, Japan), Coe comfort (GC Dental Products Corp., Tokyo, Japan) and Viscogel (Dentsply Sirona Inc., Konstanz, Germany) were used in this study. The chemical compositions are listed in Table 1 [13].

### 2.3. Microbial Culture

*C. albicans* (ATCC 10231^TM^, Manassas, VA, USA) and *S. mutans* (ATCC 25175^TM^, Manassas, VA, USA) were cultured on Sabouraud dextrose agar (Becton, Dickinson and Company, Sparks, MD, USA) and brain heart infusion agar (Becton, Dickinson and Company, Sparks, MD, USA), respectively. *C. albicans* and *S. mutans* were subcultured in Sabouraud dextrose media and brain heart infusion media, respectively, at 37 °C for 24 h before testing.

### 2.4. Antimicrobial Activity of LCEO

The antimicrobial activity of *LC*EO on *C. albicans* and *S. mutans* was assessed by using the agar disc diffusion assay. Inoculum suspensions were standardized by adjusting the turbidity to 1.0 and 0.5 McFarland standards using a McFarland densitometer (Biosan Medical-Biological Research & Technologies, GIBTHAI Co., Ltd., Bangkok, Thailand) for *C. albicans* and *S. mutans*, respectively. Then, 100 µL of diluted inoculum was used to form a lawn culture. *LC*EO was serially diluted in dimethyl sulfoxide (DMSO, Sigma-Aldrich, St. louis, MO, USA) to obtain a final concentration of 0.675, 1.25, 2.5, 5 and 10% *v*/*v*. The negative control was DMSO, and the positive control was nystatin oral suspension and 5% *v*/*v* of 2% chlorhexidine gluconate for *C. albicans* and *S. mutans*, respectively. Then, paper filter discs with a diameter of 6 mm were filled with 20 µL of the tested conditions. The plates were inoculated at 37 °C for 48 h. A digital caliper was used to measure the inhibition zone. The average inhibition zones were computed. Each experimental group (*n* = 5) was replicated five times independently.

### 2.5. Antimicrobial Activity of LCEO Incorporated into Soft Lining Materials

The antimicrobial activity of *LC*EO incorporated into GC soft liner, Coe comfort and Viscogel was assessed by using the agar well diffusion assay. Different concentrations of *LC*EO were added into the liquid components of soft lining materials before the powder was added and mixed. The final concentrations of the *LC*EO in the soft lining materials were 5%, 10%, 20% and 30% *v*/*v*. The final concentration of positive control in the materials against *S. mutans* was 5% *v*/*v* of 2% chlorhexidine gluconate (CHX), while soft lining materials with 30% *v*/*v* nystatin oral suspension (NYS) were used as a positive control against *C. albicans*. The negative control was soft lining materials without additives. All specimens were incubated at 37 °C for 48 h; the inhibition zone of each specimen was measured by using a digital veneer caliper, and the mean inhibition zones were calculated. Each experimental group (*n* = 5) was replicated five times independently.

### 2.6. Cytotoxicity Test

The test of the effects of different additives incorporated into GC soft liner, Coe comfort and Viscogel on a human gingival fibroblast (HGF) cell line (ATCC^®^ CRL-2014, Manassas, VA, USA) was adapted from ISO 10993-5 [14]. The specimens were fabricated in accordance with ISO 10993-12 [15]. A cuboidal sample, being 20 × 5 × 2 mm (3 cm^2^ /mL extracted ratio), was created with a stainless-steel mold. Soft lining materials without additives and soft lining materials with 10%, 30% *v*/*v LC*EO, NYS and CHX were tested. All specimens were exposed to UV light for 15 min per side. Subsequently, each sample was immersed in a microcentrifuge tube containing 1.0 mL Dulbecco’s modified Eagle’s medium high-glucose (Hyclone^TM^, South logan, UT, USA) with 10% fetal bovine serum and 1% antibiotics penicillin–streptomycin. The samples were incubated in a shaking incubator at 37 °C for 24 h. Culture medium without sample was also incubated under the same conditions to serve as a negative control. HGF cells were cultured in DMEM supplemented with 10% *v*/*v* FBS and 1% *v*/*v* penicillin–streptomycin solution. HGF cells were plated into 96-well plates with the seeding density of 10,000 HGF cells/well/100 μL and incubated in a 5% CO_2_ incubator at 37 °C. After 24 h, old medium was discarded, and 200 μL aliquots of prepared extracted solutions were subsequently added into the 96-well plate, and cells were incubated for 24 h. Cell viability was assessed by using the MTT assay (Sigma-Aldrich, St. louis, MO, USA). The absorbance was read at 570 nm using a microplate spectrophotometer (μQuant^TM^, Bio-Tek instruments Inc, Winooski, VT, USA). Each experimental group (*n* = 3) was replicated three times independently.

### 2.7. Shore AO Hardness Testing

Surface hardness was evaluated using a Shore durometer according to ISO 10139-1:2018 [16]. Cylindrical-shaped specimens were prepared from GC soft liner, Coe comfort and Viscogel in a stainless-steel mold with 55 mm diameter and 8.0 mm thickness [17]. Various concentrations of the additives were mixed into the soft lining materials to acquire final concentrations of 5%, 10%, 20%, 30% *v*/*v LC*EO, 5% *v*/*v* of 2% CHX and 30% *v*/*v* NYS. Soft ling materials without additive were used as controls. The mold was immersed into a 37 ± 1 °C water bath within 15 min after mixing for 2 h, and then the Shore AO hardness was measured. Each value was read after 5 s of loading at five different points and the mean hardness was calculated in each sample [17]. After this, the specimen was immersed in the water bath at 37 ± 1 °C for 7 days. The Shore AO hardness test was performed at the opposite site of the samples in the same manner. Five specimens in each group (*n* = 5) were tested.

### 2.8. Statistical Analysis

All statistical analyses were performed using PASW Statistics software (IBM Corp. Released 2019. IBM SPSS Statistics for Windows, Version 26.0, Armonk, NY, USA). The significance level was set at *α* = 0.05.

To assess the susceptibility of *S. mutans* and *C. albicans* towards *LC*EO incorporated in different soft lining materials, one-way ANOVA and Games–Howell multiple comparison were performed to analyze the significant differences in the agar well diffusion assay of *LC*EO.

To assess the HGF cell viability after 24 h exposure to the extracts, one-way ANOVA was performed for the GC soft liner and Coe comfort groups and Kruskal–Wallis analysis was used for Viscogel groups.

For the surface hardness testing, two-way repeated-measures ANOVA was applied.

## 3. Results

### 3.1. Antimicrobial Activity of LCEO towards C. albicans and S. mutans

The antimicrobial activity of *LC*EO was demonstrated as the mean inhibition zone, as listed in Table 2 and Table 3. The inhibitory zone against *C. albicans* was noticeably observed in the 1.25% *v*/*v LC*EO group, whereas the inhibitory zone against *S. mutans* was detected in the 10% *v*/*v LC*EO group (Figure 1). Similar to the NYS group, the 5% *v*/*v LCEO* group showed a remarkable inhibitory effect against *C. albicans*.

### 3.2. Antimicrobial Activity of LCEO Incorporated into Different Soft Lining Materials against C. albicans and S. mutans

The antimicrobial activity of soft lining materials with various concentrations of *LC*EO is illustrated in Figure 2. The mean inhibition zones against *C. albicans* and *S. mutans* are shown in Table 4 and Table 5, respectively. All three soft lining materials with 10% *v*/*v LC*EO displayed anti-candidal activity towards *C. albicans*. With a higher concentration of *LC*EO, the inhibition zones significantly increased in all materials (*p* < 0.05). Moreover, the inhibition zones of the groups of the three soft lining materials with 30% *v*/*v*
*LC*EO demonstrated significantly larger values than the 30% *v*/*v* NYS groups (*p* < 0.05). Meanwhile, all three soft lining materials with 5–20% *v*/*v* of *LC*EO did not show any inhibitory effect against *S. mutans.* Only in the 30% *v*/*v*
*LC*EO group, the inhibition zone could be detected around the samples. However, when compared to the CHX positive control groups, the inhibition zones of the 30% *v*/*v*
*LC*EO groups were significantly smaller (*p* < 0.05).

### 3.3. Cell Viability of HGF Cell Line

As shown in Figure 3, the cell viability test was performed to obtain the percentages of cell viability of the HGF cell line after exposure to the three soft lining materials with additives (10%, 30% *v*/*v LC*EO, CHX and NYS) compared to their control materials (no additive). Within the same material, no cell cytotoxicity was detected when the groups of various additives were compared to the control group (*p* > 0.05).

### 3.4. Surface Hardness of Soft Lining Materials

The hardness results were illustrated in terms of Shore AO hardness values (Table 6, Table 7 and Table 8), with a higher value indicating greater hardness. Overall, both the aging process and the concentrations of *LC*EO could significantly affect the Shore AO hardness of these three soft lining materials (*p* < 0.05), as shown in Figure 4. In all groups, including controls, surface hardness increased from the initial stage (2 h) to the late stage (7 days). Furthermore, the Shore AO hardness of the material was shown to decrease with increasing *LC*EO concentration. Briefly, 10%, 20% and 30% *v*/*v LC*EO incorporated into soft lining materials markedly decreased hardness in both stages (*p* < 0.05).

Nevertheless, when various additives were added, the surface hardness of each soft lining material showed different results. In the case of the GC soft liner, all additives could lower the material’s hardness at the 2 h initial stage. At the late stage, only 10% to 30% *v*/*v* of *LC*EO could significantly affect the hardness (*p* < 0.05) but no significant difference in surface hardness among control, NYS, CHX and 5% *v*/*v LC*EO was found (*p* > 0.05). Regarding Viscogel, in both the 2 h and 7 day incubation periods, there was no significant difference in surface hardness among the control, NYS, CHX and 5% *v*/*v LC*EO groups (*p* > 0.05). However, the higher concentrations from 10% to 30% *v*/*v* of *LC*EO could significantly reduce the hardness (*p* < 0.05) at 2 h and 7 days. Lastly, regarding Coe comfort, only CHX had no effect on the material’s hardness at either incubation time, while NYS and all *LC*EO concentrations could reduce the surface hardness at both 2 h and 7 days.

## 4. Discussion

Citral, the main component of fruit-derived essential oil, is the active compound of *LC*EO and was used as an antimicrobial agent against *C. albicans* and *S. mutans* in the present study. In another study, citral showed antimicrobial properties against Gram-negative and Gram-positive bacteria, fungi and protozoa, and the authors found that Gram-negative bacteria were the most resistant strains, requiring a high minimum inhibitory concentration (MIC) of approximately 18,000 μg/mL to inhibit *Pseudomonas aeruginosa* growth. Meanwhile, a lower MIC of 750 μg/mL worked against Enterobacter species [9]. A previous study also found that the MIC of 0.05% *v*/*v* citral could inhibit the growth of *S. aureus*, *E**. coli*, *C. albicans* and *Microsporum gypseum* [18]. Leite et al. revealed that citral had an MIC of 64 μg/mL in a test with all *C. albicans* strains, while the minimum fungicidal concentration (MFC) of citral ranged from 64 to 256 μg/mL depending on the candida strain [19]. In the present study, the MICs of *LC*EO against *C. albicans* and *S. mutans* were 1.25% *v*/*v* (11.1 mg/mL) and 10% *v*/*v* (89 mg/mL), respectively. The MIC of *LC*EO against *C. albicans* was in a range consistent with the results of Saikia et al. [20]. They used an agar disc diffusion experiment and found that the MIC of *LC*EO extracted from fruit against *C. albicans* ranged from 2.5 to 20 mg/mL. Conversely, Yang et al. reported that the minimum microbicidal concentration of *LC*EO from fruit against *S. mutans* was 0.75 mg/mL [21]. Therefore, variations in the type and amount of active ingredients extracted from different *L. cubeba* sources could affect *LC*EO’s antimicrobial property.

When incorporating *LC*EO into all soft lining materials, the minimum concentration of *LC*EO against *C. albicans* and *S. mutans* was 10% and 30% *v*/*v*, respectively, indicating that *LC*EO incorporated into soft lining materials could inhibit *C. albicans* more effectively than *S. mutans*. However, higher concentrations of *LC*EO were required to inhibit these oral microbes when incorporated into soft lining materials. The findings are similar to a previous study that tested *Melaleuca alternifolia* oil loaded into various tissue conditioners against *C. albicans*. A higher concentration of *M. alternifolia* oil was required to inhibit the fungus [22]. This would be because denture soft lining materials are considered to be a drug delivery vehicle. The polymer network of the soft lining materials could release active ingredients into the environment in a sustainable manner [23].

As previously stated, the incorporation of *LC*EO into soft lining materials may affect the material properties or the gingival tissue surrounding them. The in vitro cytotoxicity of GC soft liner, Viscogel and Coe comfort that contained various additives towards HGF cell lines was evaluated using the MTT assay on 24 h eluate extracts. No significant cytotoxicity towards the HGF cell line was found in any additive group, including nystatin and chlorhexidine. Correspondingly, Luo et al. [24] investigated the acute and genetic toxicity of *LC*EO in the mouse and Sprague-Dawley (SD) rat, reporting a negative result for *LC*EO genetic toxicity both in vitro and in vivo. The oral LD_50_, dermal LD_50_ and inhalation LC_50_ were listed as 4000 mg/kg, 5000 mg/kg and 12,500 ppm, respectively. The essential oil was classified as Class 5 in the globally harmonized system of classification and labeling of chemicals (GHS), which is the least toxic class [24]. However, some studies have found that high concentrations of chlorhexidine, nystatin and *LC*EO may contribute to unfavorable cell viability. A previous study, for example, reported a less than 6% survival rate of human fibroblasts, myoblasts and osteoblasts when exposed to more than 0.02% chlorhexidine gluconate [25]. Furthermore, nystatin was shown to be cytotoxic in hamster buccal epithelial cells. The study found that cells exposed to 100–200 µg/mL (0.01–0.02%) nystatin solution had a lower survival rate. Cells exposed to nystatin suspension, on the other hand, had a higher survival rate. The difference in survival rate was due to nystatin’s insolubility in water [26]. Moreover, vapor of *LC*EO induced apoptosis and cell cycle arrest in human non-small cell lung cancer cells (NSCLC cells, A549), according to a recent study [27].

In addition, incorporating substances into the polymer network chains of materials may affect their hardness. This present study revealed that when compared to the control groups, all additives incorporated into soft lining materials lowered Shore AO hardness. The higher concentration of *LC*EO resulted in lower Shore AO hardness both in 2 h and 7 day incubation periods, which indicates that adding *LC*EO also softened the materials in a dose-dependent manner. One possible factor associated with the hardness of materials would be a plasticizing effect. A plasticizer is an additive added to polymers to soften or make them more pliable. A good plasticizer is a compound that has a high degree of solvent power, is similar to the polymer chain in size and shape, as well as polarity, and can be retained within the polymer chain [28]. Therefore, the solubility parameter of polymers and additives is necessary to address the plasticizing property. The additives with a similar solubility parameter to the polymer chain have a better plasticizing effect. To evaluate the compatibility of polymers with plasticizers, Hansen’s solubility parameter was used and these parameters of tested materials are listed in Table 9 [29,30,31,32]. Since citral has solubility parameters close to polymers, *LC*EO could potentially act as a plasticizer to reduce the hardness of soft lining materials. Furthermore, not only the plasticizer concentration but also the molecular weight could affect the plasticizer’s efficiency [28]. A high-molecular-weight plasticizer has lower solubility parameters, resulting in less plasticizer efficacy. For instance, comparing 30% *v*/*v* nystatin oral suspension to 30% *v*/*v LC*EO as plasticizers, nystatin has a higher molecular weight than *LC*EO, so the Shore AO hardness of the 30% *v*/*v LC*EO group was lower than that of the 30% NYS group (*p* < 0.05).

The addition of the PMMA polymer to the powder also greatly influenced the plasticizer leachability of soft lining materials. Previous studies found that when the PMMA polymer concentration was less than 10% by weight, the leaching of the plasticizer decreased as the PMMA polymer concentration increased. The soft lining materials containing 10% by weight PMMA polymer had the lowest amount of plasticizer leaching. Meanwhile, PMMA concentrations greater than 15% by weight may result in increased plasticizer leaching out because PMMA may impede the penetration of external unreacted plasticizer [33]. According to our findings, the Viscogel powder composition contains PMMA, resulting in lower hardness changes than observed in GC soft liner and Coe comfort.

According to ISO:10139-1:2018, soft and extra-soft lining materials are classified by using Shore AO hardness after the aging process. After 2 h incubation, the hardness of a soft material should be between 30 and 50 units, and it should not exceed 60 units after 7 days. Meanwhile, the hardness of an extra-soft material should be less than 30 units after 2 h incubation and no more than 60 units after 7 days [16]. As a result, adding oil as an antimicrobial additive could lower the Shore AO hardness of the materials in the current study, but the hardness was still in accordance with ISO:10139-1.

However, the main limitation of this in vitro investigations was that, in the oral cavity, there are many other factors affecting the antimicrobial properties. Furthermore, in clinical situations, the biocompatibility of materials of significant concern. For example, saliva acts as a rinsing agent, and this may impact the release of compounds from materials, or the materials themselves may have adverse effects on the tissue underneath when using them for a period of time. Therefore, further research into the release profiles of leaching products over time and their cytotoxicity should be conducted, and a clinical study may be more beneficial.

## 5. Conclusions

Based on this in vitro study, *LC*EO could possibly be used as a novel antimicrobial additive incorporated into soft lining materials against oral pathogens, with an optimal concentration at 10% *v*/*v* for *C. albicans* and 30% *v*/*v S. mutans.* The soft lining materials with *LC*EO demonstrated no cytotoxicity towards the HGF cell line compared to materials without additives. The surface hardness of materials was dose-dependently related to the oil concentration; nonetheless, all investigated materials were classified as extra-soft types according to ISO 10139-1: 2018. Therefore, *LC*EO is a medicinal herb that has potential to treat denture stomatitis and reduce secondary caries, with no harmful effect when incorporated into denture soft lining materials.

## Figures and Tables

**Figure 1 polymers-14-03261-f001:**
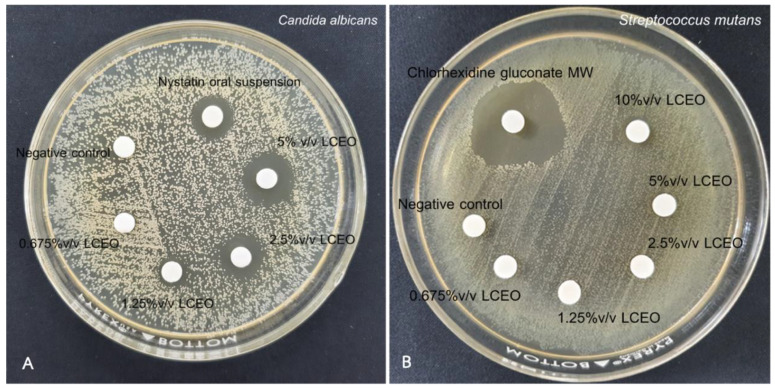
Agar disc diffusion assay of various concentrations of LCEO, positive controls (5% *v*/*v* of 2% chlorhexidine gluconate mouthwash and nystatin oral suspension) and negative control (DMSO) against *C. albicans* (**A**) and *S. mutans* (**B**).

**Figure 2 polymers-14-03261-f002:**
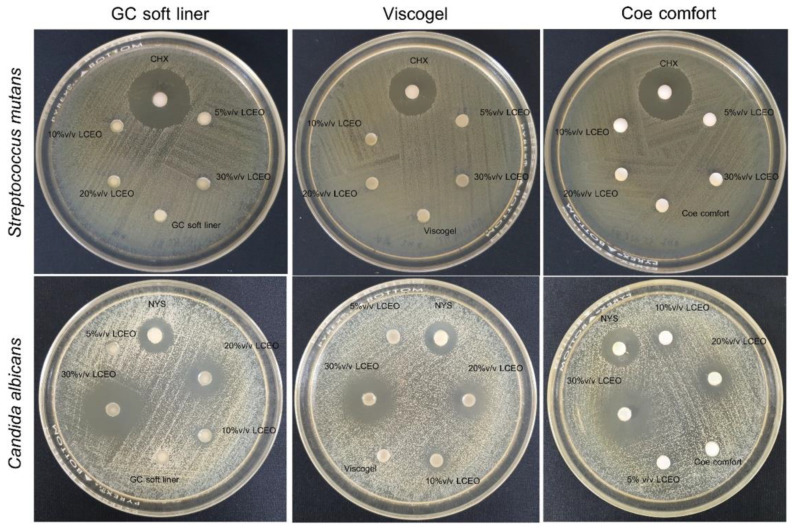
Inhibition zones of various concentrations of LCEO incorporated into three soft lining materials against *S. mutans* (**upper row**) and *C. albicans* (**lower row**). All agar well diffusion plates composed of GC soft liner (**left**), Viscogel (**middle**) and Coe comfort (**right**) incorporated with 5, 10, 20 and 30% *v*/*v* LCEO, positive control (5% *v*/*v* of 2% chlorhexidine gluconate (CHX) for *S. mutans* and 30% *v*/*v* nystatin oral suspension (NYS) for *C. albicans*) and negative control (soft lining material without additive).

**Figure 3 polymers-14-03261-f003:**
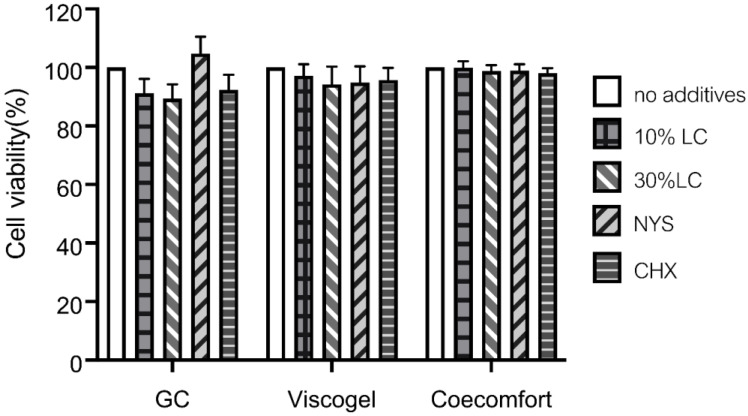
Cell viability (%) of GC soft liner, Viscogel and Coe comfort incorporated with various additives compared to soft lining materials without additive. No statistical difference was found among the same soft lining materials (*p* > 0.05). Note: 5% *v*/*v* of 2% chlorhexidine gluconate (CHX) and NYS 30% *v*/*v* nystatin oral suspension (NYS).

**Figure 4 polymers-14-03261-f004:**
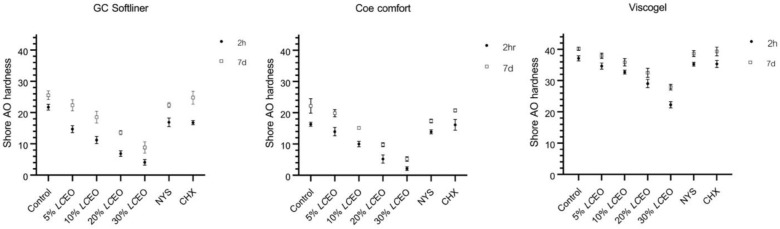
Shore AO hardness values (mean ± SD) of GC soft liner, Coe comfort and Viscogel with and without additives are presented for both 2 h and 7 day incubation times. All three soft lining materials performed evenly, with more incubation time resulting in an increase in Shore AO hardness (*p* < 0.05). When adding higher concentration of *LC*EO, the surface hardness decreased (*p* < 0.05).

**Table 1 polymers-14-03261-t001:** Chemical compositions of soft lining materials used in this study.

Product	Powder Compositions	Liquid Compositions	P/L
GC Soft liner (GC DENTAL PRODUCTS CORP., Japan)	Poly(ethyl methacrylate), PEMA (100%)	Butyl phthalyl butyl glycolate, BPBG (80.9%) Dibutyl phthalate, DBP (4.3%) Ethanol, EtOH (14.8%)	2.2 g/1.8 g
Coe comfort (GC DENTAL PRODUCTS CORP, Japan)	Poly(ethyl methacrylate), PEMA (100%)	Benzyl Benzoate, (50–70%) Ethanol (5–10%) Peppermint oil (menthol) (<1%) Butylated hydroxytoluene (<0.5%)	6 g/5 mL
Viscogel (DENTSPLY SIRONA INC., Germany)	Poly(ethyl methacrylate), PEMA (86.2%) Poly(methyl methacrylate), PMMA (13.8%)	Butyl phthalyl butyl glycolate, BPBG (86.9%) Dibutyl phthalate, DBP (8.2%) Ethanol, EtOH (4.9%)	3 g/2 mL

**Table 2 polymers-14-03261-t002:** Mean inhibition zone ± SD (*n* = 5) of different concentrations of *LC*EO against *C. albicans.* Inhibition zone less than 6 mm was defined as not detected (ND).

Condition	*Candida albicans* Mean Inhibition Zone ± SD (mm)
Negative control (DMSO)	ND
Nystatin oral suspension	12.74 ± 0.27
0.675% *v*/*v LC*EO	ND
1.25% *v*/*v LC*EO	8.33 ± 0.20
2.5% *v*/*v LC*EO	10.74 ± 0.70
5% *v*/*v LC*EO	14.33 ± 0.52

**Table 3 polymers-14-03261-t003:** Mean inhibition zone ± SD (*n* = 5) of different concentrations of *LC*EO against *S. mutans.* Inhibition zone less than 6 mm was defined as not detected (ND).

Condition	*Streptococcus mutans* Mean Inhibition Zone ± SD (mm)
Negative control (DMSO)	ND
5% *v*/*v* of 2% Chlorhexidine gluconate	21.15 ± 0.35
0.675% *v*/*v LC*EO	ND
1.25% *v*/*v LC*EO	ND
2.5% *v*/*v LC*EO	ND
5% *v*/*v LC*EO	ND
10% *v*/*v LC*EO	9.56 ± 0.52

**Table 4 polymers-14-03261-t004:** Mean inhibition zone ± SD (*n* = 5) of various additives incorporated into three soft lining materials (GC soft liner, Viscogel and Coe comfort) against *C. albicans*. Inhibition zone less than 6.0 mm was defined as not detected (ND). The statistical difference in various concentrations of *LC*EO and nystatin oral suspension is compared in the same row and the same capital letters indicate no significant difference in inhibition zone diameter (*p* > 0.05). No comparison among the three soft lining materials was performed.

Condition	Mean Inhibition Zone (Mean ± SD)					
	No Additive	5% *v*/*v LC*EO	10% *v*/*v LC*EO	20% *v*/*v* *LC*EO	30% *v*/*v* *LC*EO	30% Nystatin Oral Suspension
GC soft liner	ND	ND	7.61 ± 0.24 ^A^	13.19 ± 0.66 ^B^	22.42 ± 0.97 ^C^	12.14 ± 0.25 ^B^
Viscogel	ND	ND	12.09 ± 0.57 ^D^	16.56 ± 0.88 ^E^	24.68 ± 1.07 ^F^	11.33 ± 0.31 ^D^
Coe comfort	ND	ND	10.22 ± 0.81 ^G^	14.07 ± 0.60 ^H^	22.61 ± 0.52 ^I^	11.75± 0.37 ^G,H^

**Table 5 polymers-14-03261-t005:** Mean inhibition zone ± SD (*n* = 5) of various additives incorporated into three soft lining materials (GC soft liner, Viscogel and Coe comfort) against *S. mutans.* Inhibition zone less than 6.0 mm was defined as not detected (ND). Statistical difference in various concentrations of *LC*EO and chlorhexidine mouthwash is compared in the same row and the same capital letter*s* indicate no significant difference in inhibition zone diameter (*p* > 0.05). No comparison among the three soft lining materials was performed.

Condition	Mean Inhibition Zone (Mean ± SD)					
	No Additive	5% *v*/*v LC*EO	10% *v*/*v LC*EO	20% *v*/*v LC*EO	30% *v*/*v* *LC*EO	CHX
GC Soft liner	ND	ND	ND	ND	7.89 ± 0.40 ^A^	22.90 ± 0.75 ^B^
Viscogel	ND	ND	ND	ND	7.96 ± 0.45 ^C^	20.06 ± 0.20 ^D^
Coe comfort	ND	ND	ND	ND	8.15 ± 0.25 ^E^	20.08± 1.67 ^F^

**Table 6 polymers-14-03261-t006:** Shore AO hardness values of GC soft liner with and without additives are shown as mean ± SD (*n* = 5). Same uppercase superscript letters denote no significant difference (*p* > 0.05) among additives only within the same column. Same lowercase superscript letters denote no significant difference between 2 h and 7 days (*p* > 0.05).

Soft Lining Material/ Additives	GC Soft Liner	
	2 h	7 days
No additive	21.72 ± 0.92 ^A,a^	25.56 ± 1.39 ^G,b^
NYS	16.88 ± 1.38 ^B,a^	22.44 ± 0.82 ^G,b^
CHX	16.80 ± 0.65 ^B,a^	24.80 ± 2.08 ^G,b^
5% *v*/*v* *LC*EO	14.72 ± 1.12 ^B,a^	22.36 ± 1.79 ^G,b^
10% *v*/*v* *LC*EO	11.24 ± 1.16 ^C,a^	18.52 ± 1.89 ^H,b^
20% *v*/*v* *LC*EO	6.88 ± 0.88 ^D,a^	13.64 ± 0.75 ^I,b^
30% *v*/*v LC*EO	4.08 ± 0.88 ^E,a^	8.84 ± 1.81 ^J,b^

**Table 7 polymers-14-03261-t007:** Shore AO hardness values of Viscogel with and without additives are shown as mean ± SD (*n* = 5). Same uppercase superscript letters denote no significant difference (*p* > 0.05) among additives only within the same column. Same lowercase superscript letters denote no significant difference between 2 h and 7 days (*p* > 0.05).

Soft Lining Material/ Additives	Viscogel	
	2 h	7 days
No additive	37.13 ± 0.81 ^A,a^	40.20 ± 0.53 ^G,b^
NYS	35.27 ± 0.61 ^AB,a^	38.67 ± 0.95 ^GH,b^
CHX	35.33 ± 1.10 ^AB,a^	39.33 ± 1.40 ^G,b^
5% *v*/*v LC*EO	34.67 ± 0.99 ^AB,a^	37.93 ± 0.83 ^GH,b^
10% *v*/*v* *LC*EO	32.73 ± 0.64 ^B,a^	35.87 ± 1.17 ^H,b^
20% *v*/*v* *LC*EO	29.07 ± 1.33 ^C,a^	32.53 ± 1.45 ^I,b^
30% *v*/*v* *LC*EO	22.27 ± 0.99 ^D,a^	27.93 ± 0.90 ^J,b^

**Table 8 polymers-14-03261-t008:** Shore AO hardness values of Coe comfort with and without additives are shown as mean ± SD (*n* = 5). Same uppercase superscript letters denote no significant difference (*p* > 0.05) among additives only within the same column. Same lowercase superscript letters denote no significant difference between 2 h and 7 days (*p* > 0.05).

Soft lining Material/ Additives	Coe Comfort	
	2 h	7 days
No additive	16.32 ± 0.67 ^A,a^	22.20 ± 2.32 ^G,b^
NYS	13.88 ± 0.67 ^B,a^	17.36 ± 0.59 ^I,b^
CHX	16.12 ± 1.74 ^AB,a^	20.76 ± 0.52 ^GH,b^
5%v/v *LC*EO	13.96 ± 1.34 ^B,a^	19.88 ± 1.15 ^H,b^
10%v/v *LC*EO	10.00 ± 0.87 ^C,a^	15.16 ± 0.38 ^I,b^
20%v/v *LC*EO	5.20 ± 1.31 ^D,a^	9.76 ± 0.62 ^J,b^
30%v/v *LC*EO	2.12 ± 0.58 ^E,a^	5.20 ± 0.71 ^K,b^

**Table 9 polymers-14-03261-t009:** Summary of solubility parameters (δ, MPa^1/2^) of main composition in GC soft liner, Viscogel, Coe comfort, *LC*EO, nystatin oral suspension and chlorhexidine gluconate. ND—no documentation.

Materials	Main Composition	Hansen’s Solubility Parameter (δ, MPa^1/2^)
GC soft liner powder Coe comfort powder	Poly (ethyl methacrylate)	20.5
GC soft liner liquid	Butyl phthalyl butyl glycolate (80.9%)	20.7
	Dibutyl phthalate (4.3%)	19.0, 20.1
	Ethanol (14.8%)	26.0, 26.5
Coe comfort liquid	Benzyl Benzoate, (50–70%)	21.3
	Ethanol (5–10%)	26.0, 26.5
	Peppermint oil (menthol) (<1%)	20.2
	Butylated hydroxytoluene (<0.5%)	16.7
Viscogel powder	Poly (ethyl methacrylate) (86.2%)	20.5
	Poly (methyl methacrylate) (13.8%)	20.2, 21.5
Viscogel liquid	Butyl benzyl phthalate (87.3%)	22.4, 22.3
	Dibutyl phthalate (4.5%)	19.0, 20.1
	Ethanol (4.9%)	26.0, 26.5
*LC*EO	Citral	18.6
Nystatin oral suspension	Nystatin	ND
	Water	47.8
	Sucrose	36.3
Chlorhexidine gluconate	Chlorhexidine gluconate salt	ND
	Water	47.8

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
