# Peer review of "Effects of Litsea cubeba Essential Oil Incorporated into Denture Soft Lining Materials"

_polymers, 2022, doi:10.3390/polym14163261_

Round 1

Reviewer 1 Report

Introduction of the paper needs to be revised and supplemented,

- results shown in Table 2 and Table 3 need to be combined into one table,

- rework Figure 3  from a bar chart to another appropriate type of chart or list the results shown in Figure 3 in a table,

- combine Tables 6, 7 and 8 into one table,

- Table 9 does not belong to the Results and it is necessary to move it to the Experimental part of the paper,

- References must also be supplemented  and updated.

Reviewer 2 Report

I have received the Manuscript entitled: ‘Effects of Litsea cubeba Essential Oil Incorporated into Denture Soft Lining Materials (Manuscript ID: polymers-1856401) submitted to the Polymers for a review.

The Manuscript describes incorporation of various concentrations of Litsea cubeba essential oil (LCEO) into soft lining materials (GC soft liner, Viscogel and Coe comfort) and subsequent characterization (antimicrobial properties, cell cytotoxicity and surface hardness) of the obtained systems. The introduction section contains a well described hypothesis – the elaborated research is well designed and the upraised subjects is scientifically valuable. Authors, correctly stated that adding various substances into soft lining materials may affect their properties, which includes antimicrobial activity or mechanical durability. The methodologies developed and utilized by authors are appropriate and I could not find in this aspect any weaknesses. Moreover, the discussion in the paper is well-founded and references are cited correctly. Authors correctly selected microbial culture - C. albicans and S. mutans. Denture liners are truly prone to microbial accumulation with these microbiota. It was also established that when incorporated LCEO into all soft lining materials, the minimum concentration of LCEO against C. albicans and S. mutans was 10% and 30% v/v, respectively, indicating that LCEO incorporated into soft lining materials could inhibit C. albicans more effectively than S. mutans. I consider the reviewed Manuscript as a good candidate for publication in Polymers. Nonetheless, I recommend to correct a few minor ambiguities:

·         Lines 47-49: The sentence “In previous study, they revealed a potent growth inhibitory effect of L. cubeba on yeast and fungi (Saccharomyces cerevisiae, C. albicans, and Aspergillus nigus).” should be rephrased into “In previous study, a potent growth inhibitory effect of L. cubeba on yeast and fungi (Saccharomyces cerevisiae, C. albicans, and Aspergillus nigus)  was revealed.”

·         Line 108:  Demonstration of all units, including Celsius degrees should be unified

·         Lines 261-262: ‘Gram’ should be written starting form large letter in the whole document

·         Lines 262-263: authors wrote: “Gram-negative bacteria were the most resistant strain,...” - is there any explanation of that result?

·         Lines 356-361: Conclusions section should be strengthened with additional 2-3 sentences demonstrating the most important results and their importance in the light of available data.

Author Response

We corrected them all. Thank you so much for your suggestion.

Reviewer 3 Report

The paper entitled as ''Effects of Litsea cubeba Essential Oil Incorporated into Denture Soft Lining Materials'' by Songsang et al. is an interesting study of the natural products effects on the soft lining materials. After the incorporation of the natural oil into the soft lining materials authors have systematically tested its different effects as a function of the oil's concentration.   I really like the fact that e. g. for the hardness parameters estimation authors used 5 repetitions what makes the results they got more reliable.    Just to make this study even better, and more interesting for a broader audience of readers I would advise authors to refer to more recent papers on different materials, like potential drug delivery systems, which also have very interesting properties. Papers authors should refer to are given below:   Pharmaceutics. 2022 Apr 1;14(4):772. doi: 10.3390/pharmaceutics14040772.   and also this one about the crude plant extracts with many interesting properties in it:   Metabolites. 2022 May 17;12(5):451. doi: 10.3390/metabo12050451.  

After incorporation of this information the paper will be ready to go to the people.

Author Response

We included these two articles into the introduction section. Thank you for your suggestion.